# Targeting the DNA Damage Response Pathway as a Novel Therapeutic Strategy in Colorectal Cancer

**DOI:** 10.3390/cancers14061388

**Published:** 2022-03-09

**Authors:** Fabio Catalano, Roberto Borea, Silvia Puglisi, Andrea Boutros, Annalice Gandini, Malvina Cremante, Valentino Martelli, Stefania Sciallero, Alberto Puccini

**Affiliations:** 1Medical Oncology Unit 1, IRCCS Ospedale Policlinico San Martino, 16132 Genoa, Italy; catalan.fab@gmail.com (F.C.); roby.borea@gmail.com (R.B.); silvia.puglisi95@gmail.com (S.P.); boutros.andrea@gmail.com (A.B.); gandini.annalice@gmail.com (A.G.); malvina.cremante@gmail.com (M.C.); martellivalentino91@gmail.com (V.M.); stefania.sciallero@hsanmartino.it (S.S.); 2Department of Internal Medicine and Medical Specialties (DIMI), School of Medicine, University of Genoa, 16132 Genoa, Italy

**Keywords:** DNA damage response, genetics, clinical trial, colorectal cancer, PARP inhibitors, precision oncology

## Abstract

**Simple Summary:**

Defective DNA damage response (DDR) is a hallmark of cancer leading to genomic instability. Up to 15–20% of colorectal cancers carry alterations in DDR. However, the role of DDR alterations as a prognostic factor and as a therapeutic target must be elucidated. To date, disappointing results have been obtained in different clinical trials mainly due to poor molecular selection of patients. Several challenges must be overcome before these compounds may have an impact on colorectal cancer. For instance, although some preclinical evidence showed the vulnerability of a subset of CRCs to PARP inhibitors, no specific clinical or molecular biomarkers have been validated to select patients. Moreover, different DDR alterations may not equally confer platinum sensitivity in CRC patients. Further efforts are needed in both preclinical and clinical settings to exploit DDR alterations as therapeutic targets and to eventually discover PARP or other DDR inhibitors (e.g., Wee1) with clinical benefit on colorectal cancer patients.

**Abstract:**

Major advances have been made in CRC treatment in recent years, especially in molecularly driven therapies and immunotherapy. Despite this, a large number of advanced colorectal cancer patients do not benefit from these treatments and their prognosis remains poor. The landscape of DNA damage response (DDR) alterations is emerging as a novel target for treatment in different cancer types. PARP inhibitors have been approved for the treatment of ovarian, breast, pancreatic, and prostate cancers carrying deleterious *BRCA1/2* pathogenic variants or homologous recombination repair (HRR) deficiency (HRD). Recent research reported on the emerging role of HRD in CRC and showed that alterations in these genes, either germline or somatic, are carried by up to 15–20% of CRCs. However, the role of HRD is still widely unknown, and few data about their clinical impact are available, especially in CRC patients. In this review, we report preclinical and clinical data currently available on DDR inhibitors in CRC. We also emphasize the predictive role of DDR mutations in response to platinum-based chemotherapy and the potential clinical role of DDR inhibitors. More preclinical and clinical trials are required to better understand the impact of DDR alterations in CRC patients and the therapeutic opportunities with novel DDR inhibitors.

## 1. Introduction

Colorectal cancer (CRC) is the third most common tumor in the world [1]. CRC remains one of the principal causes of cancer-related deaths (counting ~9%), and the trend in the past 15 years shows an increase in this percentage [2]. Recently, the algorithm of treatment for advanced CRC has changed, being increasingly focused on precision medicine and on new biological drugs. For unresectable and metastatic CRC, several systemic treatment options are available, according to the mutational profile of the tumor, the type of previous treatment, and the profile of toxicities [3].

Currently, the most common treatment used in metastatic colorectal cancer (mCRC) is represented by chemotherapy regimens constituting oxaliplatin, 5-fluorouracil, and/or irinotecan in combination with targeted agents including bevacizumab or aflibercept (antiangiogenic agents) and, according to the RAS/BRAF status, cetuximab or panitumumab (anti-epidermal growth factor receptor (EGFR) drugs) [4]. Anti-epidermal growth factor receptor (EGFR) drugs such as panitumumab or cetuximab, in association with FOLFIRI or FOLFOX, are the standard first-line choice for left-sided, RAS and BRAF wildtype, and MSS mCRC [5,6]. For right-sided, RAS and BRAF wildtype, MSS mCRC, ESMO guidelines allow the use of anti-EGFR drugs in association with doublet-CT, despite the detrimental effect on OS, for shrinkage purposes in symptomatic high-volume disease [7]. In RAS-mutated tumors, regardless of left-sided or right-sided colon cancers, the association of chemotherapy with the antiangiogenic drug bevacizumab is the standard first-line treatment with a safe toxicity profile [8]. Patients harboring the *BRAF^V600E^* mutation have intrinsic resistance to anti-EGFR drugs and worse outcomes compared to BRAF wildtype tumors [9]. About 30% of *BRAF*-mutated colon cancers have an MSI-high profile; for these patients, a first-line systemic treatment with pembrolizumab is the first choice, followed by the newly approved systemic regimen containing encorafenib plus cetuximab. After a first-line therapy with doublet-CT plus bevacizumab, BRAF-mutated, MSS mCRC should also receive a second-line treatment with encorafenib + cetuximab [10]. Moreover, mCRC HER2-mutated patients may benefit from treatment with lapatinib or trastuzumab in different settings; phase 3 trials are required to confirm the preliminary antitumoral efficacy [11,12]. Moreover, RAS wildtype tumors harboring HER2 amplification may receive a second/third-line therapy with trastuzumab deruxtecan [13,14].

After the publication of the results of the phase 3 KEYNOTE-177 trial, for mismatch repair-deficient (MMR-deficient)/MSI-high mCRC, a first-line regiment with pembrolizumab has become the new standard of care [15]. Moreover, the combination of anti-PD1 (nivolumab) plus anti-CTLA4 (ipilimumab) has been tested in a first-line setting, with promising results in terms of PFS and OS in a phase 2 study [16,17].

Subsequent lines of therapy may involve the rechallenge/reintroduction of previous regimens, as well as the use of regorafenib or TAS-102 [3]. Other mutations are currently under investigation, such as *NTRK* fusion (<1% of CRC) or the *KRAS* G12C mutation (3% of mCRC), both targetable by the new drugs entrectinib and sotorasib, respectively [18,19].

An emerging aspect among different cancer types is the alteration of the DNA damage response (DDR) pathway [20]. The DDR pathway plays a key role in preserving genomic stability [21]. Many different endogenous and exogenous factors can cause genomic instability and errors during the mechanism involved in the DNA replication. Reactive oxygen species (ROS) and ionizing radiation are examples of endogenous and exogenous factors involved in the altered DNA replication [21]. The DDR pathway has already been studied in breast, ovarian, prostate, and pancreatic cancer. These tumors harboring a DDR pathway mutation benefit from platinum compounds and poly(ADP-ribose) polymerase-inhibitors (PARPi) [22,23,24]. Recent studies support the idea of the DDR pathway as an important role in the development of CRC [25,26]. A meta-analysis reported an increased risk of developing CRC in patients with a germline alteration in breast cancer gene (*BRCA*) 1, but not in *BRCA2* [27]. Different studies reported a frequency of somatic DDR mutation in CRC between 10% and 20% [28,29,30], while germline mutations have been reported in about 5% [28]. Interestingly, the frequency of DDR mutation was higher in MSI-H cancers than in MSS [31] and in right-sided, RAS wildtype, BRAF-mutant, and CMS1 subgroups [31].

Despite these advances, treatment options for the majority of mCRC patients are limited. Thus, drugs targeting *BRCA* and other DDR complex genes are being heavily investigated [32].

In this article, we review the role of the DDR pathway in CRC, highlighting available preclinical and clinical evidence to show the most promising avenues for implications of DDR alterations in CRC patients.

## 2. The DNA Damage Response (DDR) Pathway

The DDR pathway is a complex of different mechanisms including DNA damage repair, DNA damage tolerance mechanisms, and cell-cycle checkpoint control (Figure 1). This complex system regulates the proper performance of DNA replication, proliferation, and consequently, cell survival. The role of the DDR pathway is crucial in maintaining genomic integrity and stability by repairing DNA damages. Strand breakages induced by base alterations, single-strand breaks (SSBs), or double-strand breaks (DSBs) can end in chromosome breakages and, therefore, loss of genes. DNA DSBs are mostly caused by altered DNA replication forks, ROS, ionizing radiation, and physical or mechanical stress [31].

The DDR system is composed of different components: the direct reversal/repair (DR) pathway, the non-homologous end joining (NHEJ) pathway, the homologous recombination repair (HRR) pathway, the mismatch repair (MMR) pathway, the base excision repair (BER) pathway, and the nucleotide excision repair (NER) pathway [33]. The first studied gene involved in the DDR pathway was O(6)-alkylguanine-DNA methyltransferase (MGMT) [34]. MGMT fixes DNA damage by removing alkyl groups from altered impaired thymine or guanine bases. This mechanism is very important since alkylating agents create O(6)-alkylguanine in DNA, which leads to carcinogenesis [35]. The BER pathway works by removing damaged DNA bases and single-strand breaks produced by oxidation, alkylation, and deamination [36]. The MMR pathway repairs mismatches of single base pairs (A–G, T–C) and small insertion–deletion loops which are not repaired during the DNA replication S phase [37].

Among the different types of DNA damage, DSBs represent the most dangerous. The NHEJ and HRR systems are the most important mechanisms involved in neutralizing serious DNA damage. NHEJ acts throughout the cell cycle, whereas HRR is restricted to late S/G2 phases [38]. The proper balance between HRR and NHEJ is largely determined by BRCA1 and 53BP1, DDR adaptor proteins that are upregulated at DSB site [39]. The 53BP1 protein triggers the NHEJ mechanism in repairing programmed DSBs, whereas BRCA1 antagonizes 53BP1 and activates DSB resection and the HRR mechanism [40]. Whenever a two-ended DSB occurs, the NHEJ is activated, while the Ku70–Ku80 heterodimer (Ku) binds to DNA ends and recruits DNA-dependent protein kinase catalytic subunit (DNA-PKcs), creating the active DNA-PK holoenzyme [41]. On the other hand, the HRR mechanism is activated whenever the NHEJ system fails or is inappropriate. In these cases, the DSBs are exposed to 5′-end resection, generating 3′ single-stranded (ss) DNA that interferes with Ku binding and supports the repair by HRR. The RE11–RAD50–NBS1 (MRN) complex is the first one engaged in the lesion to activate the HRR mechanism. Subsequently, BRCA2, together with PALB2, BRCA1, and RAD51, creates a complex which leads to the formation of a new nucleoprotein filament [42].

The cause behind inactivation of the DDR mechanism in cancer development can be a genetic and/or an epigenetic alteration [43,44]. The most characteristic method is genetic inactivation; this mechanism can change DNA sequences by germline or somatic mutations. An example of germline inactivation is Lynch syndrome [33]. An analysis of 500 metastatic tumors showed a prevalence of pathogenic germline variants in 12.2%, 75% of which were DDR-related mutations [45]. Epigenetic instability also plays an important role in carcinogenesis. The microsatellite instability (MSI) phenotype is induced by different alterations, with the most common being epigenetic silencing of MLH1, such as MLH1 hypermethylation [46]. The complete loss of function of the NEHJ mechanism causes a high number of DSBs and a subsequent cell death impossible to prevent [47]. This is the reason why only a few cases of downregulation or alteration of core NHEJ genes have been described [47]. Nevertheless, HRR somatic alterations are the most frequent DNA repair pathway among DDR genes over 33 cancer types [48]. *BRCA1, BRCA2, RAD51, BLM,* and *RAD50* are the most common mutations associated with homologous recombination deficiency (HRD) [48].

In the last decade, the role of DNA damage repair (DDR) gene mutations has become more and more relevant; they were shown to be a positive predictive marker of sensitivity to platinum-based chemotherapy regimens and poly(ADP-ribose) polymerase inhibitor (PARPi) response in different tumors, including breast, ovarian, pancreatic, and prostate cancer [49]. The therapeutic role of platinum compounds and PARPi in *BRCA* mutated breast and ovarian cancer is well established and a standard of care [50,51]. the Food and Drug Administration (FDA) only approved the use of olaparib [52] and rucaparib [53] in 2020 for patients with DDR-mutated metastatic castration-resistant prostate cancer (mCRPC). Regarding gastrointestinal (GI) tract tumors, promising results have been reported with olaparib as maintenance therapy after platinum-based chemotherapy in patients with *BRCA1*/2-mutated pancreatic cancer (POLO trial) [22,54]. Therefore, olaparib was approved by the FDA and EMA in this therapeutic setting [55].

In CRC, only few data are available about the prognostic and predictive role of DDR alterations. However, a recent meta-analysis reported that carrying BRCA1 and/or BRCA2 mutation is not associated with a higher risk of developing CRC, in contrast with previously reported data (higher risk due to BRCA1 mutation) [27,56]. Despite that, germline alterations in *BRCA1* and *BRCA2* genes seem to be associated with early-onset CRC [57]. The risk for CRC, as well as for anal carcinoma, in *BRCA* carriers seems to be elevated in women below the age of 50 [58], and *BRCA*-mutated CRCs were often of mucinous histology [59]. This histology subtype is also associated with other defects in DNA repair genes, such as the MMR genes, suggesting a distinct tumor biology that deserves further investigation.

Interestingly, a recent article by Sayed et al. described the plausible mechanism of microbe-induced impairment of DNA repair by specific downregulation of a BER protein, NEIL2. Indeed, they showed that *Fusobacterium nucleatum* induces the downregulation of NEIL2 and accumulation of DNA damage, eventually leading to CRC progression [60]. This may represent new avenues to be investigated to develop further effective treatments for CRC patients.

## 3. Biomarkers of DDR: How to Select Patients

The identification of biomarkers of DNA damage response (DDR) genomic alterations and of predictive biomarkers of the response to PARP inhibitors is a current unmet clinical need [61]. In fact, a substantial number of patients who do not carry somatic or germline *BRCA1/2* mutations may still benefit from PARP inhibitors. Moreover, some *BRCA1/2* mutations carriers may not respond to PARP inhibitors [62,63].

Therefore, several studies investigated the molecular characteristics of patients with *BRCA1/2*-mutated tumors that could serve as biomarkers to select those *BRCA1/2* wildtype patients that could benefit from a PARP inhibitor. Hence, the term ‘BRCAness’ was coined to define those tumors showing molecular and phenotypic features similar to those found in *BRCA1/2*-mutated tumors. These characteristics can arise from a range of both genomic and/or epigenetic alterations [61].

The term ‘BRCAness’ was then expanded to ‘HRDness’ to better include non-BRCA-related sensitivity to PARP inhibitors [61]. The term ‘PARPness’ can be used when sensitivity to PARP inhibitors goes beyond the mechanisms of HRD, through different molecular alterations from base-excision repair (BER), alternative NHEJ, or replication-fork protection [64]. However, outside of these definitions, there is to date no consistent identification of patients who might respond to PARP inhibitors. The main clinical criteria of sensitivity to PARP inhibitors used in most clinical trials is responsiveness to platinum-based chemotherapy [65]. This is well defined in ovarian cancer, whereas there is still no commonly accepted definition of ‘platinum sensitivity’ in CRC.

Efforts to develop molecular biomarkers of HRD have included transcriptomic signatures [66], mutational signatures [67], BRCA1 and RAD51 promoter hypermethylation [68], and functional biomarkers. However, these approaches are not widely available in everyday clinical practice.

The main molecular approach adopted in clinical trials is the detection of genetic alterations through next-generation sequencing (NGS) technologies.

In particular, the FoundationFocus CDx _BRCA LOH_ test was developed to predict the efficacy of the PARP inhibitor rucaparib in high-grade ovarian cancer with *BRCA* alterations and/or genomic LOH, referred to as HRD (tBRCA^+^ and/or LOH_high_) [69]. It consists of three main elements: (a) germline *BRCA* gene test, (b) NGS analysis of somatic variants of HR and NHEJ pathways, and (c) SNP profiling to evaluate LOH score [70].

The ARIEL2 study, involving platinum-sensitive relapsed ovarian cancer patients treated with rucaparib, showed that LOH_high_ status (defined as ≥14%) had a stronger predictive value (78% of responders predicted) compared to other biomarkers (e.g., *BRCA1* or *RAD51* methylation or mutation in other HRR genes: 48% and 11%, respectively) [71]. The overall response rates (ORR) in the gBRCA1/2, *sBRCA1/2*, LOH_high_, and intention-to-treat population were 85%, 74%, 29%, and 31%, respectively [71].

In the ARIEL3 trial, rucaparib maintenance therapy was evaluated in the same platinum-sensitive relapsed ovarian cancer population, specifically in patients carrying a *BRCA* somatic mutation, HR-deficient (*sBRCAm* and *BRCA* wildtype LOH_high_), and HR-proficient, with a threshold to define LOH_high_ as ≥16%. Median PFS in the *sBRCAm* patients was 16.6 vs. 5.4 months in the placebo group (HR 0.23; 0.16–0.34), in the non-*BRCAm* HRD group was 13.6 vs. 5.4 months (HR 0.32; 0.24–0.42), and in the intention-to-treat population was 10.8 vs. 5.4 months (HR 0.36; 0.30–0.45) [63].

However, there are still no data regarding the prevalence of genomic scarring in CRC. In this regard, Foundation Medicine’s HRR–HRD Lynparza assay has developed a panel of 15 HR genes that could be considered in future studies investigating HRD in CRC [70].

The Myriad myChoice HRD test assesses BRCAness by global genomic scarring through (a) loss of heterozygosity (LOH), (b) telomeric allelic imbalance (TAI), and (c) large-scale state transition (LST). TAI consists of a discrepancy in the 1:1 allele ratio in the telomere, while LST consists of transition points in different regions of DNA [72]. This test was used in the NOVA study, where maintenance therapy with niraparib in platinum-sensitive recurrent ovarian cancer showed a substantial benefit in terms of PFS [62]. However, the benefit provided by the PARP inhibitor in this trial was observed independently of the HRD status [62]. Thus, even genomic scarring assays are not inclusive enough in defining the molecular signatures of HRD or in identifying the mechanisms of sensitivity to PARP inhibitors. Probably, the selection of a platinum-sensitive population in this study might have selected only HRD patients, although HRD was not detectable by the assay in some patients.

With the advent and spreading of genomic sequencing, several studies have been conducted on the identification of genomic signatures of HRDness. A study by Alexandrov and colleagues retrospectively studied all mutations associated with *BRCA1/2*-mutated tumors and identified signatures associated with *BRCA1/2*-inactivating mutations on a variety of tumors, such as the so-called ‘Signature 3’ [67]. However, this signature does not have a cutoff point to distinguish *BRCA*-proficient from *BRCA*-deficient tumors. On the basis of these data, the HRDetect assay was designed to identify BRCA-deficient tumors with a sensitivity of 98.7% [73].

However, despite their high sensitivity in detecting BRCA-deficient tumors, ‘Signature 3′ and HRDetect failed to detect other functionally relevant mutations in other HR pathways [73].

Many studies on PARP inhibitors have observed objective responses in patients with HRD tumors harboring non-BRCA-related mutations [24,64,74,75,76], such as ATM, CHEK, and ATR, as well as other genes involved in chromatin remodeling such as ARID1A [77] and BAP1 [78,79], or transcriptional regulators of DDR genes such as CDK12 [80].

The increasing use of NGS techniques has widened the range of predictive biomarkers of sensitivity to PARP inhibitors, but it must be underlined that these biomarkers only represent a snapshot of the mutational status of the tumor at the time of biopsy.

In order to have dynamic biomarkers that would better describe the evolution of tumor mutational status over time, serial monitoring of circulating free DNA (cfDNA), before, during, and at disease progression, has been investigated [81]. This has been studied for example in metastatic castration-resistant prostate cancer (mCRPC), where NGS testing on cfDNA at disease progression revealed the reversion of HR mutations, leading to a re-establishment of DDR gene function and, consequently, drug resistance [81].

In addition, other dynamic biomarkers can be identified from preclinical comparisons between HRD and HR-proficient cell lines in RNA profiling [82].

Promising alternative approaches, especially for CRC and breast cancer, in predicting the benefit from PARP inhibitors are the assessment of gammaH2AX and *RAD51* after radiation-induced DNA damage. However, this approach requires an engineered system based on plasmid transfection, making it not easily feasible on a routine basis [83].

Furthermore, tumors lacking HRDness characteristics may also be sensitive to PARP inhibitors, falling under the definition of ‘PARPness’. Potential biomarkers of PARPness include levels of PARP1, E-cadherin, and/or Schlafen 11 (SLFN11) [84,85]. Other biomarkers include those associated with high levels of cell replication, a sign of instability and replicative stress. IDH1 mutations in gliomas have been shown to confer sensitivity to PARP inhibitors by reducing the production of NAD^+^, a molecular compound required for PARP1-mediated DNA repair [86].

To address the biological rationale for a wider use of PARP inhibitors for different tumor types, there is a relevant need to go beyond the *BRCA1/2*-centered mutational landscape.

In summary, there is still no standard test or a benchmark assay for the identification of the HRD tumors that might be responsive to PARP inhibitors. In particular for CRC, further efforts are needed to increase both the efficacy and clinical feasibility of these assays to allow PARP inhibitors to enter clinical practice in CRC bearing genomic alterations in DNA damage response.

## 4. Preclinical Data in Colorectal Cancer

The efficacy of PARPi has been proven in different cancer types. So far, only a few preclinical works have assessed the efficacy of PARPi or other DDR inhibitors in CRC, especially in MSS CRC patients, which still represent an unmet clinical need.

Among genes involved in DDR, McAndrew and colleagues focused on *RAD54B*, an effector of the HR pathway. PARP1 silencing or inhibition selectively killed RAD54B-deficient cells, with a concomitant increase in γ-H2AX, which is a marker of DNA DSBs, as well as cleaved caspase-3 (an indicator of apoptosis) [87].

Wang et al. tested the same hypothesis in CRC cell lines and demonstrated that ATM^−/−^ cells with depletion of p53 have enhanced sensitivity to PARPi [88].

Other groups, as demonstrated by the work of Ozden et al. [89], investigated the role of BARD1, a protein involved in BRCA1 stabilization and functioning, in PARPi sensitivity. Assuming that an oncogenic BARD1 splicing variant may render cancer cells more sensitive to HR inhibition, they showed how, in cell lines where there were higher levels of BARD1beta, which creates an unstable BARD1/BRCA complex, there was an increased sensibility to PARPi.

PARP sensitivity may also be related to TP53 status, as described Smeby and colleagues [90]. They investigated 93 CRC cell lines to evaluate PARPi sensitivity; MSI cell lines had in general a higher PARP inhibition sensitivity index compared to MSS ones. Among MSS cell lines, *TP53* status was found to be related with PARPi sensitivity.

Other preclinical studies have investigated the interaction between PARPi and other chemotherapeutic agents.

Kaiwu et al. have studied the interaction between olaparib and oxaliplatin in one CRC line (SW480). They showed how olaparib may enhance the effect of oxaliplatin, since cells treated with the combination of the two drugs exhibit higher sensitivity to oxaliplatin (the surviving fraction of cells in response to olaparib combined with oxaliplatin was significantly lower than that of the control) and a higher fraction of cells arrested their cell cycle in G2/M phase [91].

Genther et al. studied the effect of niraparib and then the combination of niraparib + SN-38 (the active metabolite of irinotecan) in CRC cell lines stratified by MMR status [92]. The study demonstrated that MSI phenotype does not sensitize CRC cell lines to PARP inhibition, but MSI cells were more sensitive to SN-38. Similar results were found by Tahara and colleagues; in their study, they also found that this effect may be amplified in RAD51-deficient cells [93].

Augustine et al. investigated the efficacy of different PARP inhibitors and DNA-damaging agents, both as single agents and in combinations, in CRC cells [94]. The greatest synergy was demonstrated with rucaparib and irinotecan combination, followed by olaparib in combination with PJ34, especially in MSI cells and when the two agents were used concomitantly. No synergy was seen with the combination of oxaliplatin and PARPi, in contrast with previous findings.

Another important effector of HRD system is *ATM*. Greene et al. [95] tested whether an *ATM* inhibitor (AZ31) would enhance sensitivity to irinotecan in CRC cell lines and CRC patient-derived xenografts. They demonstrated that AZ31 strengthens the effect of SN-38, especially in primary resistant irinotecan cells. In addition to this, in tumors exhibiting irinotecan resistance and combination sensitivity, they found an association between *PIK3CA* mutation and combination sensitivity.

Other preclinical studies have been conducted in order to find potential biomarkers to PARPi sensitivity. Arena and colleagues tested the sensitivity to olaparib and to oxaliplatin and 5-fluorouracil (5-FU) in CRC cell lines, patient-derived organoids (PDO), and patient-derived xenografts (PDX) enriched for *KRAS* and *BRAF* mutations [83]. They found that up to 13% of them undergo growth arrest after 2 weeks of exposure to clinically achievable levels of olaparib and display functional deficiency in HR. PARPi sensitivity was positively correlated with sensitivity to oxaliplatin, and treatment with olaparib impaired tumor growth; in addition, maintenance therapy with PARP blockade after initial oxaliplatin response delayed disease progression in mice. This work, conducted on a poor-prognosis CRC subset, suggests the importance of identifying patients in whom colorectal cancer is more likely to benefit from olaparib. They also analyzed whether biomarkers predictive of clinical benefit from PARP inhibitors in other malignancies could be applied to identify colorectal cancer models responsive to olaparib. They demonstrated that genomic features associated with BRCAness or HR repair diagnostic assays do not directly correlate to PARPi sensitivity, whereas other assays based on detection of DDR were able to pinpoint vulnerability to PARP inhibition.

However, considering these results, further preclinical studies are needed to assess the efficacy of PARPi as maintenance therapy in CRC, as already approved in other malignancies.

## 5. Clinical Data and Ongoing Trials in CRC

### 5.1. PARP Inhibitors

Only few clinical data are available on the use of PARPi in CRC. Phase 1/2 studies investigated the use of PARPi in association with chemotherapy or PAPRi alone in cohorts of patients with solid malignancies, including CRCs.

However, to date, no large clinical trials targeting somatic *BRCA*-mutant CRC patients have been published (Table 1). Several phase 1 studies were conducted with PARPi in combination with other drugs or alone in patients with pretreated mCRC [96,97,98,99,100], unselected for any specific DDR gene alterations.

*BRCA* status was not assessed as an eligibility criterion in several small clinical trials investigating PARPi in CRC, and only one ongoing phase II trial ([104], NCT04171700) selected patients with solid tumors, including CRC, according to deleterious pathogenic variants in HRR genes.

A randomized phase 2 trial investigated the use of olaparib 400 mg BID as a single agent in patients with mCRC, after progression on systemic therapy (20 microsatellite stable (MSS), 13 MSI-H). In both MSI-H and MSS mCRC, olaparib failed to demonstrate activity (PFS 1.84 months; no complete or partial responses were reported) [101].

Veliparib was the first PARPi investigated in mCRC in combination with irinotecan or oxaliplatin, demonstrating a synergistic activity with standard chemotherapeutical regimens [105]. In a phase 1b trial, veliparib was also associated with capecitabine and radiotherapy with an acceptable safety profile, but preliminary antitumor activity needs confirmation on larger studies [99]. In a study combining veliparib with FOLFIRI with or without bevacizumab, an ORR of 57% was observed [102]. However, adding veliparib to standard treatment did not demonstrate differences in PFS or OS. No stratification according to BRCA-mutant status was performed.

More promising results were achieved in a single-arm phase 2 study in heavily pretreated mCRC patients (*N* = 50 + 5 patients with mismatch repair-deficient (dMMR)) with two cycles of veliparib plus temozolomide; a disease control rate (DCR) of 24% and two partial responses were reported. Moreover, PTEN and MGMT protein expressions assessed in tumor specimens were not associated with DCR. Five patients with dMMR tumors were unrolled and seemed to have the worst outcomes [103].

Ongoing trials with PARPi are listed in Table 2.

### 5.2. Not Only PARPi: Other Inhibitors of the DDR System

The DNA damage response pathway is an extremely complex system that can be split into two categories: the pathways involved in the repair of single-strand breaks (SSBs) of DNA and those involved in the repair of double-strand breaks (DSBs) of DNA; the latter is constituted by the mechanisms of homologous recombination (HR) and nonhomologous end joining (NHEJ), which include PARP proteins [61]. The HR system is an error-free pathway that can be used only in the late S and G2 phase, whereas NHEJ is an error-prone repair pathway that can occur throughout all cell-cycle phases [106].

As said in previous paragraphs, the ‘BRCAness phenotype’ is associated with patients with defects in HR family genes different from *BRCA*; these mutations are not surely known to drive carcinogenesis but can possibly cause deficiency in DNA repair [107]. This can be clinically relevant not only because loss of function in genes involved in DDR can predict sensitivity to DNA damage-inducing agents, such as platinum, but also because they can be targeted with specific inhibitors [61]. For instance, defects in *ATM* can confer susceptibility to both PARP inhibitors and oxaliplatin in CRC [108,109].

Indeed, the phosphatidylinositol 3-kinase (PI3K)-like kinase (PIKK) family has a role as *mediator* in the initiation of repair pathways. Members of this family are *ATM*, ataxia telangiectasia and Rad3-related protein (*ATR*), and DNA-dependent kinase (DNA-PK) [106].

*ATM* and *ATR* respond to different stimuli. After a DSB, the MRN complex, constituting MRE11, RAD50, and NBS1 proteins, binds the break and recruits the DDR proteins, including ATM. This protein is able to phosphorylate multiple targets facilitating cellular responses to the damage: H2AX, which recruits MDC1 and consequentially more molecules of MRN and ATM, amplifying the same system; NBS1 and Chk2, which arrest the cell cycle in the S and G2/M phase; BRCA1 and p53, which induce arrest in the G1 phase, upregulating the expression of p21. By contrast, ATR is activated when an SSB occurs, thereby activating Chk1 kinase that induces cell-cycle arrest; since SSBs can take place naturally during DNA replication, ATR is essential for survival under basal conditions [106].

The last mediator involved is DNA-PK, which, after its recruitment and activation by the complex Ku70/Ku80 DSBs, provides access to end processing enzymes, such as ARTEMIS [110], and phosphorylates XRCC4/LIG4, which promotes the re-ligation of the broken ends with the help of the stimulatory factor XLF [111]. It is noteworthy that ATM also acts on ARTEMIS, and this supports the concept that ATM, ATR, and DNA-PK have redundant roles [111].

Strictly related to mediators are the effectors, among which Chk1 and Chk2 have already been mentioned as effectors of ATR and ATM signaling, respectively [61]. Above these, Wee1 is a tyrosine kinase that prevents the entrance in the mitotic phase by inactivating CDK1 when DNA damage occurs. Therefore, the inhibition of Wee1 can sensitize tumors to DNA-damaging therapies [112,113].

Acting with drugs targeted against these specific molecules involved in the DNA repair cascade, next to PARP proteins, has a strong rationale. Clinical application is still premature, and different trials are ongoing.

The largest study is investigating the addition of elimusertib (BAY 1895344) to FOLFIRI in stomach and intestinal advanced or metastatic cancers. This is a still recruiting phase 1b study and the expected enrolment is 90 patients. Elimusertib is a specific inhibitor of *ATR*, and this class of drugs has already been studied, especially in ovarian and breast cancer; supposing that the same mechanism can be translated in CRC, they seem to be more effective when combined with genotoxic agents such as chemotherapies [107]. The primary aim of the study is to determine the safety and maximum tolerated dose, and secondary aims are the objective response rate (ORR), progression-free survival (PFS), and overall survival (OS) [114] (NCT04535401).

Another ATR inhibitor under initial investigation is ceralasertib (AZD6738); different studies, the majority of which aim to investigate safety and tolerability, have already been published, including different solid tumors, mainly melanomas [115,116], as well as gynecological, lung, and intestinal neoplasms [117]. Currently, the DASH trial is still recruiting patients to evaluate the safety of the association of ceralasertib and trastuzumab/deruxtecan in solid tumors characterized by HER2 expression [118] (NCT04704661).

Moreover, a phase 1b trial with the specific inhibitor of Wee1 adavosertib has just closed enrolment (AZD1775). This drug was administered in association to irinotecan as second line in patients affected by metastatic RAS- or BRAF-mutated CRC. The outcomes were tolerability and objective response, but few patients were included (seven), and the results are still unpublished [119] (NCT02906059). Adavosertib, in addition, seems to double PFS compared with active monitoring in mCRC with both TP53 and RAS mutations, which were stable or responding after 16 weeks of chemotherapy [120].

In conclusion, we can state that DDR-targeted therapy has a strong biological rationale in the treatment of CRC, but clinical data and applications are still immature and further investigations are needed.

## 6. Hereditary Implications

Patients affected by DDR alterations should be discussed by a molecular tumor board, in order to identify those eligible for targeted therapies together with those affected by hereditary syndromes [121].

This might be of great relevance to both patients and their relatives. In particular, for carriers of pathogenic germline variants in *BRCA* 1 and 2, personalized follow-up should be tailored for patients surviving their first cancer, in order to decrease their risk of non-CRCs. In addition, surveillance programs should be offered to their relatives at risk, to reduce their mortality for cancer [122].

For other rarer DDR-related syndromes, less consolidated guidelines are available, but multidisciplinary discussion should increasingly help selected patient referral to genetic counseling and detection of germline pathogenic variants. For example, *ATM* pathogenic germline variants are increasingly found to be associated with a wider cancer spectrum than previously known, and cancer risk management programs for *ATM* carriers have been proposed [123,124,125].

Increased DDR testing will also increase complexity, with a need for adequate interpretation of results, in particular for variants of unknown significance (VUS). For this purpose, Lorans et al. highlighted the need for international collaborations to create large databases. These will allow researchers to conduct well-powered studies to determine CRC susceptibility with a specific gene, develop better tools for the interpretation of VUS, and conduct more accessible clinical translation research [126].

## 7. Conclusions

The implementation of DDR alterations into clinical practice is one of the most promising research avenues for CRC to date. However, the exploitation of DDR defects in CRC patients is at a very early stage of development, and several challenges must be addressed in order to have novel compounds available for clinical use.

Indeed, a better understanding of DDR alterations and their impact on CRC biology is necessary. One of the most challenging issues is to understand which DDR alterations can be used as a novel biomarker. This would help to better identify new subsets of patients who more likely would benefit from PARPi or other DDRi (e.g., ATR or Wee1 inhibitors) and achieve the most benefit from platinum-based regimens in frontline treatment, as well as in the reintroduction setting.

For these reasons, we believe that further efforts in both preclinical and clinical settings with robust translational research should be carried out to make DDR alterations suitable targets for drug development and to eventually improve outcome in CRC patients.

## Figures and Tables

**Figure 1 cancers-14-01388-f001:**
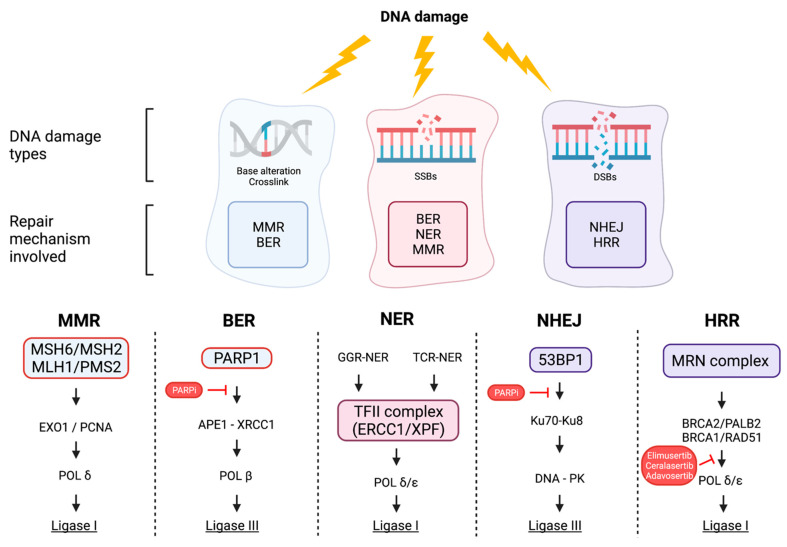
Different mechanisms for DNA damage response and main inhibitors. DNA damage is classified into different categories on the basis of the DNA damage type: single-strand breaks (SSBs), double-strand breaks (DSBs), base alterations, and crosslinks. DNA damage activates the DNA damage response pathway (DDR) that is composed of several downstream signaling pathways based on the DNA damage type. The MMR and BER pathways are activated by crosslinks, base alterations, and SSBs, the NER pathway is activated by by SSBs, and the NHEJ and HRR pathways are activated by DSBs. We also represented the main drugs and their inhibitor sites that are currently under investigation as new possible treatments in colorectal cancer (CRC). MMR, mismatch repair; BER, base excision repair; NER, nucleotide excision repair; NHEJ, nonhomologous end joining; HRR, homologous recombination repair; PARPi, PARP inhibitors.

**Table 1 cancers-14-01388-t001:** Published trials on PARPi in mCRC.

Authors/Year	Phase	Patient Population	Drugs	Results	Ref.
Leichman et al., 2016	2	CRC, 33 patients (20 MSS; 13 MSI-H)	Olaparib (AZD-2281)	No complete or partial responses were reported; ORR 0%	[101]
Gorbunova et al., 2019	2	mCRC, 130 patients	Veliparib + FOLFIRI ± bevacizumab vs. Placebo + FOLFIRI ± bevacizumab	mPFS 12 vs. 11 months mOS 25 vs. 27 months mDOR 11 vs. 9 months ORR 57%	[102]
Pishvaian et al., 2018	2	mCRC, 75 patients	Veliparib + temozolomide	DCR 24% mPFS 1.8 months mOS 6.6 months	[103]
Czito et al., 2017	1b	Locally advanced RC, 32 patients	Veliparib + capecitabine + RT	29% of patients achieved CR	[99]
Samol et al., 2012	1	mCRC, 8 patients	Olaparib + topotecan	ORR 0%	[97]
Kummar et al., 2011	1	CRC, 5 patients	Veliparib + topotecan	ORR 0%	[98]
Berlin et al., 2018	1	CRC, 10 patients	Veliparib + FOLFIRI	ORR 20%	[96]

CR, complete response; CRC, colorectal cancer; DCR, disease control rate; mCRC, metastatic colorectal cancer; mDOR, median duration of response; mOS, median overall survival; mPFS, median progression-free survival; MSI-H, microsatellite instability-high; MSS, microsatellite stable; ORR, overall response rate; RC, rectal cancer; REF, reference; RT, radiotherapy.

**Table 2 cancers-14-01388-t002:** Active clinical trials on PARPi in mCRC.

Clinical Trial	Phase	Patient Population	Mutations	Treatment Arm(s)
NCT02484404	1–2	Advanced solid tumors	Not required	MEDI4736 (anti PD1) + olaparib and/or cediranib
NCT04171700	2	Solid tumors	Deleterious mutation (germline or somatic) in BRCA1, BRCA2, PALB2, RAD51C, RAD51D, BARD1, BRIP1, FANCA, NBN, RAD51, or RAD51B	Rucaparib
NCT04166435	2	mCRC	MGMT promoter hypermethylation	Temozolomide + olaparib
NCT03983993	2	mCRC	Not required (RAS wildtype)	Niraparib + panitumumab
NCT04456699	3	mCRC	Not required	Olaparib OR olaparib + bevacizumab Vs. bevacizumab + 5-FU
NCT04511039	1	CRC or gastroesophageal cancer	Not required	Trifluridine and tipiracil hydrochloride + talazoparib *
NCT03337087	1–2	Advanced pancreatic, colorectal, gastroesophageal, or biliary cancer	Only for pancreatic cancer: BRCA1 or BRCA2 or PALB2 mutation, or HRD (non-BRCA, non-PALB)	Rucaparib + liposomal irinotecan + fluorouracil + leucovorin calcium
NCT03842228	1	Advanced solid tumors	Germline or somatic mutations in DDR genes (*ARID1A, ATM, ATRX, BARD1, BRCA1, BRCA2, BRIP1, CDK12, CHEK1, CHEK2, FANCA, FANCL, MRE11A, MSH2, PALB2, PARP1, POLD1, PP2R2A, RAD51B, RAD51C, RAD51D, RAD54L, or XRCC2*), actionable mutations in the *PTEN* gene, or hotspot mutations in the *PIK3CA* gene (E542, E545, or H1047)	Olaparib + MEDI4736 (durvalumab) + copanlisib hydrocloride
NCT04123366	2	Advanced solid tumors	Known or suspected deleterious mutations in ≥1 of the specified 15 genes involved in HRR	Olaparib + pembrolizumab
NCT04497116	1–2	Advanced solid tumors	ATR inhibitor-sensitizing mutations	RP-3500 (oral ATR inhibitor) ± talazoparib
NCT03127215	2	Advanced solid tumors	Defective DNA repair via HRR	Trabectedin/olaparib vs. physician’s choice
NCT04276376	2	Advanced solid tumors	In CRC cohort: *ATM, BARD1, BRCA1, BRCA2, BRIP1, CDK12, CHEK2, PALB2, RAD51C, RAD51D, FANCA, NBN, RAD51, RAD54L*	Rucaparib + atezolizumab
NCT03851614	2	mCRC or Pancreatic adenocarcinoma or Leyomiosarcoma	Not required (CRC patients must have MMR proficiency disease)	Olaparib + durvalumab OR cediranib + durvalumab *
NCT04693468	1	Metastatic solid tumors	Defect in DDR, *MET, ALK,* or *ROS1* genes	Talazoparib + palbociclib OR talazoparib + axitinib OR talazoparib + crizotinib
NCT03772561	1	Advanced solid tumors	Not required	AZD5363 + olaparib + durvalumab
NCT04672460	1	Advanced solid tumors	Solid tumors with known or likely pathogenic germline or somatic variants in *BRCA1* or *BRCA2* that would benefit from PARPi therapy	Talazoparib capsule vs. talazoparib soft gel capsule

5-FU, 5-fluorouracil; CRC, colorectal cancer; DDR, DNA damage response, HRR, homologous recombination repair; HRD, homologous recombination deficiency; mCRC, metastatic colorectal cancer; MMR, DNA mismatch repair; PARPi, poly(ADP-ribose) polymerase inhibitors. * Active, recruiting.

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
