# Peer review of "Targeting the DNA Damage Response Pathway as a Novel Therapeutic Strategy in Colorectal Cancer"

_cancers, 2022, doi:10.3390/cancers14061388_

Round 1

Reviewer 1 Report

This review paper by Catalano et al. is a nice and timely venture to organize the preclinical and clinical data currently available on DNA Damage Response inhibitors in colorectal cancer. CRC is a major cause of death among the world population and development of effective mechanism based therapeutic strategy remains elusive till date. In this context, this review is extremely well organized and discussed all the major facets of the current state of art in the field. The authors provide a brief introduction about the current known mechanisms of CRCs, followed by an overview of the various DNA damage response pathways operative in mammalian cells. Following this, they have provided insight into the biomarkers of DDR and the parameters to select patients based on the biomarker selection. Then the authors elaborated on the preclinical trials and current ongoing trials in CRC therapeutics with a special emphasis of various DDR inhibitors like the PARP inhibitors. They have presented their data in a tabular format which is an extremely useful summary of work.

This reviewer compliments the huge efforts by the authors to make this possible and the choice of the topics.

The reviewer has a suggestion. Since they have focused principally on the DNA repair defect in CRCs, they may cite a recent paper in Cells entitled "The DNA Glycosylase NEIL2 suppresses Fusobacterium -infection -induced inflammation and DNA damage in colonic epithelial cells". This study described the plausible mechanism of microbe-induced impairment of DNA repair by specific downregulation of a Base Excision Repair protein, NEIL2. Since NEIL2 also possesses anti-inflammatory properties as shown by Tapryal et al. Journal of Biological Chemistry (2021 Jan-Jun;296:100723. doi: 10.1016/j.jbc.2021.100723. Epub 2021 Apr 28.PMID: 33932404), thus this also augments the inflammation in the colonic epithelial cells. 

Also, they can in general discuss about the microbe-associated CRCs and any possible therapeutic options for them with focus to DDR modulators.

Reviewer 2 Report

The review study of Catalano and colleagues are advanced in the field, very well written, and well updated across contexts. There is a minor concern that needs to be addressed before the article can be accepted. Overall English language and style are acceptable but minor spell check is required.

1. What is the main question addressed by the research?
• Little is known about exploiting DDR defects in CRC. In this review study, Catalano and colleagues propose DDR transformation into clinical practice as one of the most promising research avenues to date for CRC. Thus, these authors suggested better identifying new subsets of patients more possible to have PARPi or other DDRi (eg, ATR or WEE1 inhibitors) and achieve the most usefulness from platinum-based regimens in frontline treatment in CRC. Hence, Catalano and colleagues presented a better understanding of DDR changes in CRC biology through the efforts in both preclinical and clinical stages with robust translational research that should be carried on to make DDR alterations appropriate targets for drug action in CRC patients.
2. Do you consider the topic original or relevant in the field? Does it address a specific gap in the field?
• Yes, the topic is original and relevant in the field and addresses a specific gap in the field. I believe this review would be very useful for the clinical perspective in CRC.
3. What does it add to the subject area compared with other published material?
• This review provides new insight into DDR alterations in clinical practice in CRC.
4. What specific improvements should the authors consider regarding the
methodology? What further controls should be considered?
• The methodology is fine and no further control is required.
5. Are the conclusions consistent with the evidence and arguments presented and do the
address the main question posed?
• I found the conclusion to be in line with the evidence and arguments presented, and yes, the authors address the main question beautifully.
6. Are the references appropriate?
• The references are well updated.
7. Please include any additional comments on the tables and figures.
• Tables and figures are fine.
